

# Biomedical literature classification using encyclopedic knowledge: a Wikipedia-based bag-of-concepts approach

Marcos Antonio Mouriño García, Roberto Pérez Rodríguez and Luis E. Anido Rifón

Department of Telematics Engineering, University of Vigo, Vigo, Spain

## ABSTRACT

Automatic classification of text documents into a set of categories has a lot of applications. Among those applications, the automatic classification of biomedical literature stands out as an important application for automatic document classification strategies. Biomedical staff and researchers have to deal with a lot of literature in their daily activities, so it would be useful a system that allows for accessing to documents of interest in a simple and effective way; thus, it is necessary that these documents are sorted based on some criteria—that is to say, they have to be classified. Documents to classify are usually represented following the bag-of-words (BoW) paradigm. Features are words in the text—thus suffering from synonymy and polysemy—and their weights are just based on their frequency of occurrence. This paper presents an empirical study of the efficiency of a classifier that leverages encyclopedic background knowledge—concretely Wikipedia—in order to create bag-of-concepts (BoC) representations of documents, understanding concept as "unit of meaning", and thus tackling synonymy and polysemy. Besides, the weighting of concepts is based on their semantic relevance in the text. For the evaluation of the proposal, empirical experiments have been conducted with one of the commonly used corpora for evaluating classification and retrieval of biomedical information, OHSUMED, and also with a purpose-built corpus of MEDLINE biomedical abstracts, UVigoMED. Results obtained show that the Wikipedia-based bag-of-concepts representation outperforms the classical bag-of-words representation up to 157% in the single-label classification problem and up to 100% in the multi-label problem for OHSUMED corpus, and up to 122% in the single-label classification problem and up to 155% in the multi-label problem for UVigoMED corpus.

Corresponding author
Marcos Antonio Mouriño García,
marcosmourino@gmail.com

## INTRODUCTION

The ability to automatically classify text documents into a predefined set of categories is extremely convenient. Examples of this include the classification of educational resources

into subjects such as mathematics, science, or history; the classification of books into thematic areas; and the classification of news into sections such as economy, politics, or sports. Among these and other applications, the automatic classification of biomedical literature stands out as an important application to leverage text document automatic classification strategies. Medical staff, scientists, and biomedical researchers handle in their daily work huge amounts of literature and biomedical information, so it is necessary to have a system that allows for accessing to documents of interest in a simple, effective, efficient, and quick way; thus saving time querying or searching for these documents. This implies the necessity for sorting or ranking the documents based on some criterion, i.e., their classification.

Classification is modelled as a supervised learning problem: first, the classifier is trained with a certain number of examples—documents whose category is known—and then, the algorithm is applied to another set of documents whose category is unknown (*Sebastiani, 2002*). There is a huge amount of classification algorithms, including *k*-Nearest Neighbor (KNN), Decision Tree (DT), Neural Networks, Bayes, and Support Vector Machines (SVM) (*Yang, 1999*).

The functioning of classifiers is based on the application of Natural Language Processing (NLP) techniques to the documents to classify, so that a software agent can recognise which category a given document belongs to, based on some NLP feature contained in it, such as word occurrence frequency or the structure of the language used (*Settles, 2010*). Vector Space Model (VSM) (*Salton, Wong & Yang, 1975*) is the most often used representation, where each document within a collection is represented as a point in space, commonly using as weights the frequency of occurrence of words. When words are used as features, the model is known as bag-of-words, being a bag—or multiset—a set of elements that can occur more than once (*Blizard, 1988*). Then, by using this representation, a document is characterised by a set of words that appear in text, repeated as many times as occurrences in text.

Despite being one of the traditionally used representations in document classification tasks (*Täckström, 2005*), the BoW model is suboptimal, because it only accounts for word frequency in the documents, and ignores important semantic relationships between them (*Wang et al., 2008*). The main limitations of the BoW representation are redundancy, ambiguity, orthogonality, hyponymy and hypernymy problems, data sparseness and word usage diversity. Redundancy means that synonymous are not unified (*Egozi, Markovitch & Gabrilovich, 2011*; *Huang & Milne, 2012*). For instance, in the BoW model, if a document that contains the word "tumour" was classified into the "cancer" category, this would not provide information to classify a document that contains the phrase "neoplasm".

- Ambiguity refers to the problem of polysemy—one word can have several meanings (*Täckström, 2005*; *Egozi, Markovitch & Gabrilovich, 2011*). For instance, in the BoW model, if a document that contains the word "tissue" was classified into the "human anatomy" category, it would may cause errors when classifying a document that contains the word "tissue", but meaning the *Triphosa dubitata* moth.

- Orthogonality problem means that the semantic relatedness between words is not taken into account (*Huang & Milne, 2012*). For example, knowing that a document that contains "cardiovascular system" was classified under the label "circulatory system" would not give information on how to classify a document that contains the word "blood".

- Hyponymy and hypernymy problem means that the hierarchical relations are not leveraged (*Levelt, 1993*; *Wang et al., 2007*; *Wang et al., 2008*). For instance, if a document that contains the word "heart" was classified into "human body" category, this would not provide information to classify a document that contains the word "organ" and vice versa.

- BoW representations often suffer from the problems of data sparseness (zero-probability problem) and word usage diversity. This is because the BoW model only considers frequencies of words occurring in a class, and each document often contains only a small fraction of all words in the lexicon, which degrades the performance of the classifer (*Täckström, 2005*; *Tsao, Chen & Wang, 2013*).

The most relevant works found in the literature focus mainly on solving two of the aforementioned problems: synonymy and the polysemy. To accomplish this, several authors have proposed a concept-based document representation, defining concept as "unit of meaning" (*Medelyan, Witten & Milne, 2008*; *Wang et al., 2008*; *Stock, 2010*). Several previous works demonstrated that this representation provides good results in classification tasks (*Sahlgren & Cöster, 2004*; *Wang et al., 2008*).

The literature hosts several ways to create this based-of-concepts representation. In Latent Semantic Analysis (LSA) (*Deerwester et al., 1990*; *Landauer & Dumais, 1997*) a concept is a vector that represents the context in which a term occurs; this approach overcomes synonymy but not polysemy. In Latent Dirichlet Allocation (LDA) (*Blei, Ng & Jordan, 2003*) each concept consists of a bag-of-words that represents an underlying topic in the text. In Explicit Semantic Analysis (ESA) (*Gabrilovich & Markovitch, 2007*) concepts are entries from external knowledge bases such as Wikipedia, WordNet, or Open Directory Project (ODP); these concepts are assigned to documents—annotation process—in accordance with its overlap with each entry in the knowledge base; its main disadvantage is its tendency toward generating outliers (*Egozi, Markovitch & Gabrilovich, 2011*)—concepts that have a weak relationship to the document to annotate. Semantic annotators—the approach used in our proposal—extract concepts, disambiguate them, link them to domain-specific external sources—such as Unified Medical Language System (UMLS) or Medical Subject Headings (MeSH)—or to general-purpose external sources—such as Wikipedia—and deal with synonymy and polysemy problems.

We think that there is a research gap in the application of BoC representations that leverage encyclopedic knowledge in the building of classifiers of biomedical literature. This article aims at bridging this gap by designing, developing, and evaluating a classifier—single-label and multi-label—of biomedical literature that builds on encyclopedic knowledge and represents documents as bags-of-concepts. In order to evaluate the system,

we conducted several experiments with one of the most commonly used corpora for evaluating classification and retrieval of biomedical information—OHSUMED—as well as with a purpose-built corpus that comprises MEDLINE biomedical abstracts published in 2014—UVigoMED. Results obtained show a superior performance of the classifier when using the BoC representation, and it is an indicative of the potential of the proposed system to automatically classify scientific literature in the biomedical domain.

The remainder of this article is organised as follows: 'Background' presents some background knowledge; 'Materials and Methods' presents the corpora used, the algorithms, classification strategies, and metrics employed, and the approach proposed; 'Results' shows results obtained; 'Discussion' discusses the results obtained and presents proposals for future work; finally, 'Conclusions' presents the conclusions obtained.

## BACKGROUND

In order to create the BoC representation of documents we use a general purpose semantic annotator. The literature contains other proposals for the creation of representations as bags-of-concepts. In this section, we discuss the main proposals for creating representations of documents as bags-of-concepts—Latent Semantic Analysis, Latent Dirichlet Allocation, Explicit Semantic Analysis, domain-specific semantic annotators, general-purpose semantic annotators, and hybrid semantic annotators—and proposals for biomedical literature classification that make use of these representations.

### Latent Semantic Analysis

In the theoretical basis of Latent Semantic Analysis model underlies the distributional hypothesis (*Harris, 1968*; *Sahlgren, 2008*): words that appear in similar contexts have similar meanings (*Deerwester et al., 1990*; *Landauer & Dumais, 1997*). In LSA, the meaning of a word is represented as a vector of occurrences of that word in different contexts—being a context a text document. Although LSA combats the synonymy problem, it does not combat polysemy.

The LSA model has been used by several authors for biomedical literature classification tasks. *Kim, Howland & Park (2005)* explore the dimensionality reduction provided by LSA for classifying a subset of MEDLINE, reporting precision values reaching 90%. *Täckström (2005)* also makes use of the LSA model for the categorisation of a subset of MEDLINE, obtaining positive results using BoC in categories where BoW fails; despite the fact that results are positive, the author recommends using BoW as the primary representation mechanism and BoC as a punctual complement.

### Latent Dirichlet Allocation

Latent Dirichlet Allocation model (*Blei, Ng & Jordan, 2003*) presupposes that each document within a collection comprises a small number of topics, each one of them "generating" words. Thus, LDA automatically finds topics in a text, or in other words, LDA attempts "to go back" from the document and find the set of topics that may have generated it. *Zheng, McLean & Lu (2006)* make use of LDA to identify biological topics—i.e., concepts—from a corpus composed of biomedical articles that belong to MEDLINE;

to that end, first, they use LDA to identify the most relevant concepts, and subsequently, these concepts are mapped to a biomedical vocabulary: Gene Ontology. *Phan, Nguyen & Horiguchi (2008)* get good results in the classification of short texts—OSHUMED abstracts—making use of a BoC document representation whose concepts were extracted using LDA. *Zhang, Phan & Horiguchi (2008)* focus on improving the performance of a classifier, making use of LDA to reduce the dimensionality of the set of features employed; the proposed method is applied to the biomedical corpus OHSUMED, obtaining results that demonstrate that the approach proposed provides better precision values, while reducing the size of the feature space.

## Explicit Semantic Analysis

*Gabrilovich & Markovitch (2007)* propose Explicit Semantic Analysis, a technique that leverages external knowledge sources—as Wikipedia or ODP—to generate features from text documents. Contrary to LSA and LDA, ESA makes textual analysis identifying topics that are explicitly present in background knowledge bases—such as Wikipedia or ODP, among others—instead of latent topics. In other words, ESA analyses a text to index it with Wikipedia concepts. *Gabrilovich & Markovitch (2009)* use ESA to extract features from a text and to classify text documents from MEDLINE in categories. Authors report improvements in classification performance by using ESA to generate features of documents.

## Semantic annotators

A semantic annotator is a software agent that is responsible for extracting the concepts that define a document, linking or mapping these concepts to entries from external sources. Semantic annotators usually perform disambiguation, thus combating synonymy and polysemy problems; and, in some cases, they assign a weight to each extracted concept in accordance with its semantic relevance within the document. Depending on the external source employed to link or map the extracted concepts, two kinds of semantic annotators can be distinguished: domain-specific semantic annotators and general-purpose semantic annotators.

### Domain-specific semantic annotators

Domain-specific semantic annotators use external sources of a particular domain as knowledge bases to map extracted concepts. In the biomedical domain there are several biomedical ontologies, being the most relevant in the state-of-the-art MeSH (*Lowe & Barnett, 1994*; *Lipscomb, 2000*) and UMLS (*Bodenreider, 2004*). We can find several domain-specific semantic annotators in the literature. *Elkin et al. (1988)* propose a tool to identify MeSH terms in narrative texts. *Aronson (2001)* describes the MetaMap program, which embeds an algorithm that allows for representing biomedical texts through UMLS concepts. *Jonquet, Shah & Musen (2009)* present Open Biomedical Annotator: first, it extracts terms from text documents making use of Mgrep (*Dai et al., 2008*); second, it maps these terms to biomedical concepts from UMLS and other biomedical ontologies from the National Centre from Biomedical Ontologies (NCBO); and, finally, it annotates

the documents with these concepts. *Kang et al. (2012)* combine seven domain-specific annotators—ABNER, Lingpipe, MetaMap, OpenNLP Chunker, JNET, Peregrine and StandforNer—to extract medical concepts from clinical texts, providing better results than any of the individual systems alone. Several authors make use of these and other semantic annotators for biomedical classification tasks such as: *Yetisgen-Yildiz & Pratt (2005)*, who use MetaMap to extract concepts from documents and use it to classify biomedical literature; and *Zhou, Zhang & Hu (2008a)*, who use a semantic annotator based on UMLS (MaxMatcher (*Zhou, Zhang & Hu, 2006*)) for the Bayesian classification of the biomedical literature corpus OHSUMED.

### General-purpose semantic annotators

General-purpose semantic annotators use generic knowledge bases—they are not specific of a particular domain—such as Wikipedia, WordNet of FreeBase instead of domain-specific ontologies. *Vivaldi & Rodríguez (2010)* present a system to extract concepts from biomedical text using Wikipedia as semantic information source, and *Huang & Milne (2012)* propose the use of a semantic annotator—using Wikipedia and WordNet as knowledge bases—for creating BoC representations from documents and their use in biomedical literature classification tasks.

### Hybrid semantic annotators

Hybrid semantic annotators use domain-specific ontologies—such as UMLS or MeSH—and generic knowledge bases—such as WordNet—as background knowledge to extract concepts from a narrative text. Thus, they leverage the advantages of both approaches—the specificity provided by domain-specific ontologies and the generality provided by generic knowledge bases. *Bloehdorn & Hotho (2004)* use this technique to enrich BoW representations of texts with concepts extracted from the text itself making use of MeSH ontology and the lexical database WordNet. This enriched representation is then used to perform the classification of the biomedical literature corpus OHSUMED, reporting F1-values of 48%.

## MATERIALS AND METHODS

### Dataset

#### OHSUMED

In order to evaluate the proposed system, we conducted four experiments with the well-known corpus for information retrieval and classification tasks OHSUMED. To carry out the experiments with the multi-label classifier, we used a subset of OHSUMED composed of 23,166 biomedical abstracts of 1991, classified into one or several of the 23 possible categories (*Joachims, 1998*). In order to create train and test sequences, we randomly split the corpus in a training sequence that comprises 18,533 documents and a test sequence composed of the remaining 4,633 documents.

To perform the single-label experiments, we removed from the aforementioned corpus those documents belonging to more than one category, resulting in a corpus formed by 9,034 documents classified in only one of the 23 categories; and, then, we randomised it
again to split it in a training sequence composed of 7,227 documents and a test sequence that comprises 1,807 documents.

### UVigoMED

In order to corroborate the results obtained when conducting the experiments over OHSUMED corpus, we expressly created another corpus to conduct the same experiments as in OHSUMED. We named it UVigoMED.[1] In this section, we briefly describe the corpus and the process of collecting documents. First, we selected the classification scheme, consisting of the MeSH general terms of "diseases" group—the same as in OHSUMED. It is worth noting that, to create the UVigoMED corpus, we used the 2015 MeSH tree structure, where the diseases group contains 26 categories instead of the 23 that contained the MeSH tree structure when OHSUMED was created. To build the corpus we performed the following steps (see Fig. 1):

- We downloaded from MEDLINE all the descriptions of the articles (HTML webpages) of year 2014 classified under each one of the 26 categories.
- We extracted from each article description: the title, the abstract, and the categories it belongs to.
- We stored in our database the title, abstract and categories for each article description that was downloaded.

As a result, we obtained a corpus that comprises 92,661 biomedical articles classified in one or several categories of the 26 that were available. Finally, in order to create the training and test sequences, we randomly selected 18,532 documents as the test sequence, remaining 74,129 for the training sequence.

To carry out the single-label experiments, we created a subset of the aforementioned corpus comprising those documents belonging to just one category—by removing those that belonged to more than one category—resulting in a corpus composed of 54,853 documents classified in one of the 26 categories, and split randomly in a training sequence that comprises 43,882 documents and a test sequence composed by 10,971 items.

## Multi-label classification methods

There are two main approaches to the multi-label classification problem: problem transformation methods and algorithm adaptation methods. Problem transformation methods are those that transform the multi-label problem in several single-label problems, whereas algorithm adaptation methods consist in performing adaptations of specific algorithms to address multi-label problems directly without performing any transformation.

In our proposal, we opted for using the methods of the first category, i.e., transforming the multi-label problem in N single-label binary problems, one for each category. To perform this, we made use of *Scikit-learn*, a module for Python that provides a set of the most relevant machine learning algorithms in the state-of-the-art (*Pedregosa et al., 2012*). In particular, we made use of the *one-vs-rest* or *one-vs-all* strategy, that automatically implements a classifier for each category. This strategy also allows for using different classification algorithms, including SVM, which is the one what we chose.

[1] Corpus is available at http://www.itec-sde.net/UVigoMED.zip.

## SVM algorithm

Support Vector Machines are a set of supervised machine learning algorithms used in clustering, regression, and classification tasks, among others. We selected the SVM algorithm because it is one of the most relevant algorithms in the state-of-the-art—together with Naïve Bayes, $k$-Nearest Neighbor, Decision Trees, or Neural Networks,— it is one of the most successful machine learning algorithms to perform automatic text classification tasks (*Rigutini, Maggini & Liu, 2005*) and it offers higher performance than other relevant algorithms of the state-of-the art such as KNN or Naïve Bayes (*Yang, 1999*). Although a more detailed definition can be found in *Hearst et al. (1998)*, the basic idea is that, given a set of items belonging to a set of categories, SVM builds a model that can predict which category the new items that appear in the system belong to. SVM represents each item as a point in space, separating the categories as much as possible. Then, when a new item appears in the model, it will be placed in one category or another, depending on their proximity to each one. This algorithm corresponds to the class *sklearn.svm.LinearSVC* of the *Scikit-learn* library.

## Evaluation metrics

The single-label and multi-label classification problems make use of different evaluation metrics. Hereafter, we cite the main metrics that the literature shows to evaluate each of the problems.

### Single-label classification problem

When predicting the category to which a document belongs, there are four possible outcomes: true positive (TP), true negative (TN), false positive (FP) and false negative (FN), where *positive* means that a document was classified in a certain category, *negative* means the opposite, *true* means that the classification was correct and *false* means that the classification was incorrect (*Sahlgren & Cöster, 2004*).

In the same way as *Sebastiani (2002)* and *Sahlgren & Cöster (2004)*, we define:

$$P = \text{Precision} = \frac{TP}{(TP + FP)} \tag{1}$$

$$R = \text{Recall} = \frac{TP}{(TP + FN)}. \tag{2}$$

We also use a measure that combines precision and recall, F1-score, defined as:

$$F_1 = \frac{2 * P * R}{P + R}. \tag{3}$$

In our work we report the results as macro-F1, because it is the best metric to reflect the classification performance in corpora where data are not evenly distributed over different categories (*Zhou, Zhang & Hu, 2008a*).

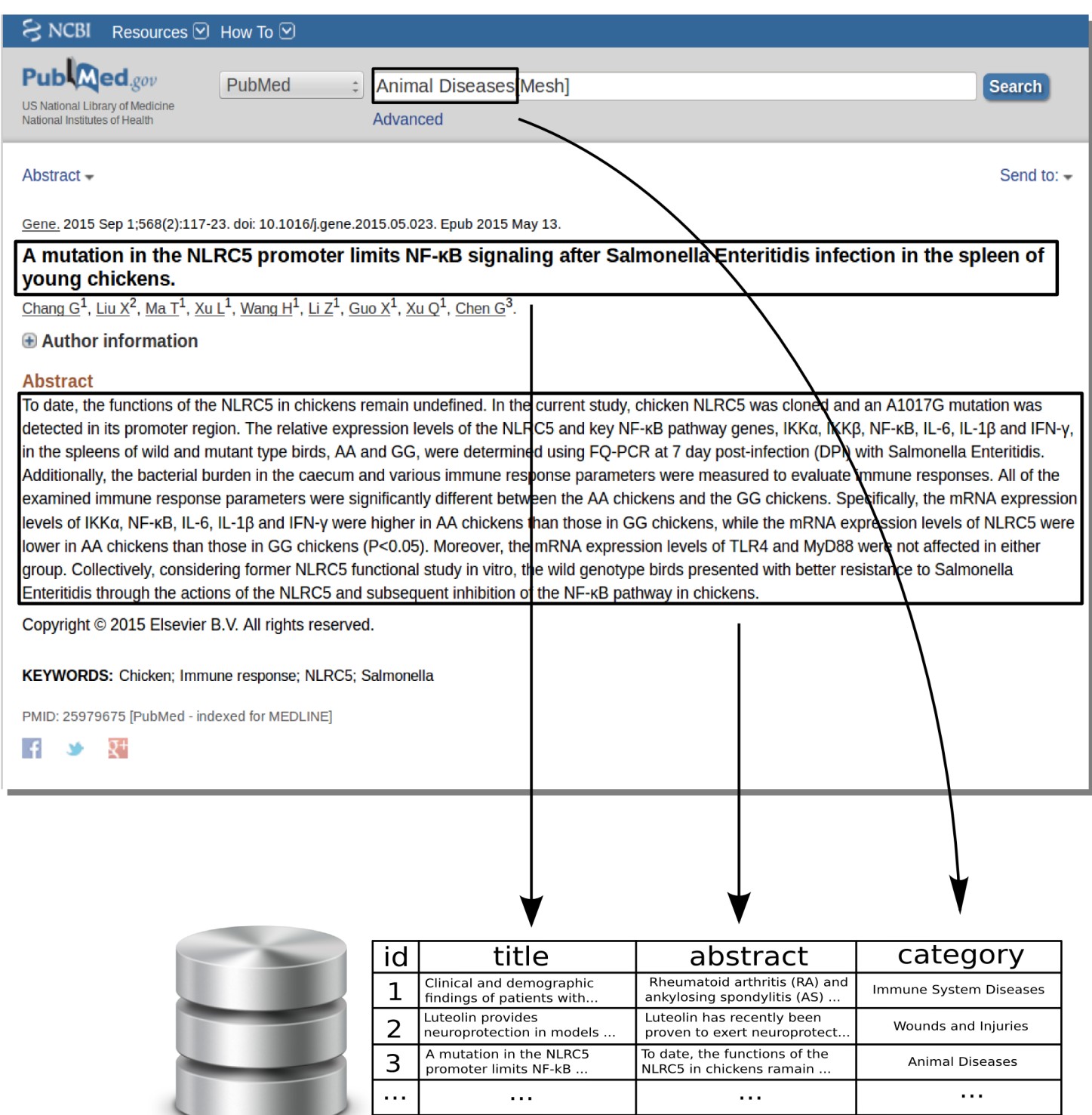

**Figure 1  UVigoMED corpus creation.**

### Multi-label classification problem

*Schapire & Singer (2000)* consider in their work the *Hamming Loss*, defined according to *Tsoumakas & Katakis (2007)* as

$$HL = \text{Hamming Loss}(H, D) = \frac{1}{|D|} \sum_{i=1}^{|D|} \frac{|Y_i \Delta Z_i|}{|L|} \tag{4}$$

being $D$ the multi-label corpus, that comprises $|D|$ multi-label elements $(x_i, Y_i), i = 1...|D|$, $Y_i \subseteq L$, $L$ is the set of labels, composed by $|L|$ labels, $H$ is a multi-label classifier, $Z = H(x_i)$ is the set of labels predicted by $H$ for $x_i$, and $\Delta$ represents the symmetric difference between two sets, corresponding to the XOR operation in the Boolean algebra.

The following metrics—*Accuracy, Precision, and Recall*—are used by *Godbole & Sarawagi (2004)* and defined again by *Tsoumakas & Katakis (2007)* as:

$$A = \text{Accuracy}(H, D) = \frac{1}{|D|} \sum_{i=1}^{|D|} \frac{|Y_i \cap Z_i|}{|Y_i \cup Z_i|} \tag{5}$$

$$P = \text{Precision}(H, D) = \frac{1}{|D|} \sum_{i=1}^{|D|} \frac{|Y_i \cap Z_i|}{|Z_i|} \tag{6}$$

$$R = \text{Recall}(H, D) = \frac{1}{|D|} \sum_{i=1}^{|D|} \frac{|Y_i \cap Z_i|}{|Y_i|}. \tag{7}$$

We also use the F1-score, defined in the previous section.

## Approach

The approach presented consists in the classification—single-label and multi-label—of the two corpora of biomedical literature defined in 'Dataset' using a Wikipedia-based bag-of-concepts representation of documents, and the comparison of the performance with the performance of the classifier when using the traditional BoW representation of documents. We used the SVM algorithm ('SVM algorithm'), and for the multi-label problem, we also made use of the strategy presented in 'Multi-label classication methods'. With the aim of conducting all the experiments under the same conditions, we selected randomly for both corpora—single-label and multi-label versions—training sequences composed of 5,000 elements and test sequences that comprise 1,000 elements.

First, it was necessary to obtain the BoW and BoC representations of each document in the corpora. Figure 2 shows the differences between the creation of the traditional BoW representation and the BoC representation. In order to create the BoW representation of a document, the first step is to filter the stop words. Stop words are words such as "the", "if", and "or" that are of no use for text classification, since they probably occur in almost all documents. The next step is stemming, the removing of common inflexional affixes, in order to perform some form of morphological normalization to create more general features. To that end we use the *Porter stemmer* (*Porter, 1980*), which is the most common

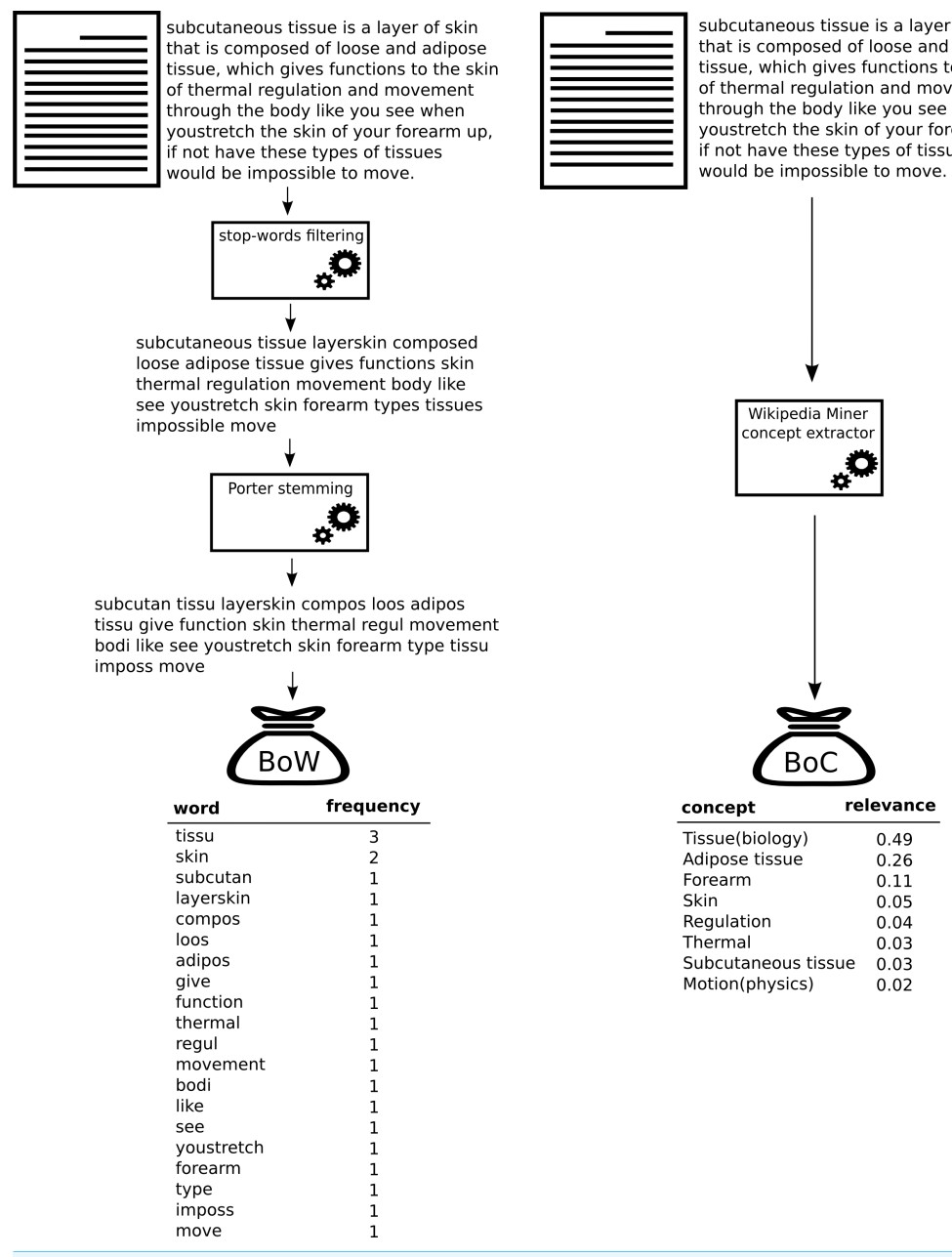

**Figure 2** Bag-of-words and bag-of-concepts creation process.

stemming algorithm to work with English text (*Täckström, 2005*). Finally, we calculate the frequency of occurrence of stemmed words.

To create the BoC, we opted for using a semantic annotator, in particular, a general-purpose semantic annotator that uses NLP techniques, machine learning, and Wikipedia as a knowledge base: *Wikipedia Miner* (*Milne & Witten, 2013*). The implementation of its algorithm is based on three steps:

- The first step is *candidate selection*. It consists in, given a text document composed of a set of *n-grams*—being an *n-gram* continuous sequence of n words—the algorithm

queries a vocabulary that comprises all the *anchor texts* in Wikipedia and verifies whether any of the *n-grams* are present in the vocabulary. Thus, for each matching *n-gram-anchor text* a candidate is obtained, being the most relevant candidates those that are most frequently used as *anchor texts* in Wikipedia.

- The second step is *disambiguation*. Given the same vocabulary of *anchor texts*, the algorithm selects the most suitable target for each candidate. The process is performed making use of machine learning techniques and using Wikipedia articles as the training sequence, since they contain good examples of manually performed disambiguation. Disambiguation is accomplished having into account the relationship of each candidate with other non-ambiguous terms in its context, and also the commonness of the candidate.

- The third step is *link detection*, wherein the relevance of concepts extracted from the text is calculated. To that end, the algorithm uses again machine learning techniques and Wikipedia articles as the training sequence, since each of them is a good example of what constitutes a relevant link and what does not. Figure 3 shows graphically the whole process to obtain a bag-of-concepts—being each one of them a Wikipedia article—from a text document.

Having obtained the BoC representation for each of the documents we proceeded to classify the two corpora—both single-label and multi-label versions—making use of the strategies and algorithms defined in 'Multi-label classication methods' and 'SVM algorithm'.

## RESULTS

### OHSUMED

Figure 4 and Table 1 show the evolution of the F1-score for BoW and BoC, varying the length of the training sequence for the single-labelled OHSUMED corpus; and Fig. 5 and Table 2, show the F1-score for BoW and BoC, varying the length of the training sequence in the multi-labelled OHSUMED corpus. We can perceive that the performance offered by the classifier using the BoC representation is clearly superior to the one offered by the traditional BoW representation for both experiments—single-label and multi-label. As we can see in Fig. 8, the BoC representation reaches improvements up to 157% for the single-label problem and up to 100% for the multi-label problem.

### UVigoMED

Figure 6 and Table 3 show the evolution of the F1-score for BoW and BoC when varying the length of the training sequence for the single-label version of the UVigoMED corpus. We can see that the performance offered by the classifier when using the BoC representation is much higher than that offered when using the BoW one, reaching improvements up to 122%, as shown in Fig. 8. The experiments conducted with the multi-label corpus provide the results shown in Fig. 7 and Table 4, where we can see again that the BoC representation outperforms BoW, reaching increases up to 155%.

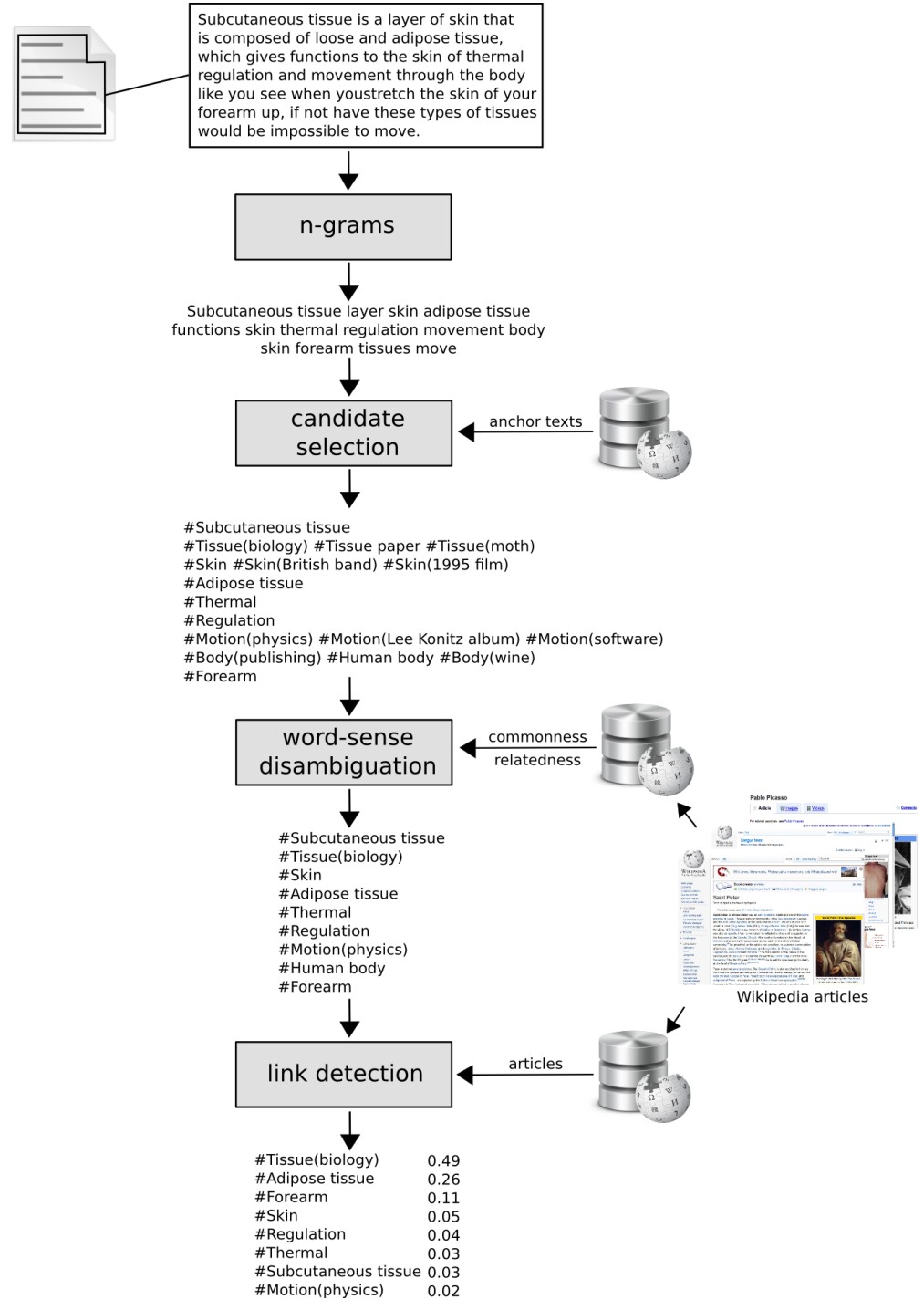

**Figure 3   Bag-of-concepts obtainment process of a document using Wikipedia Miner.**

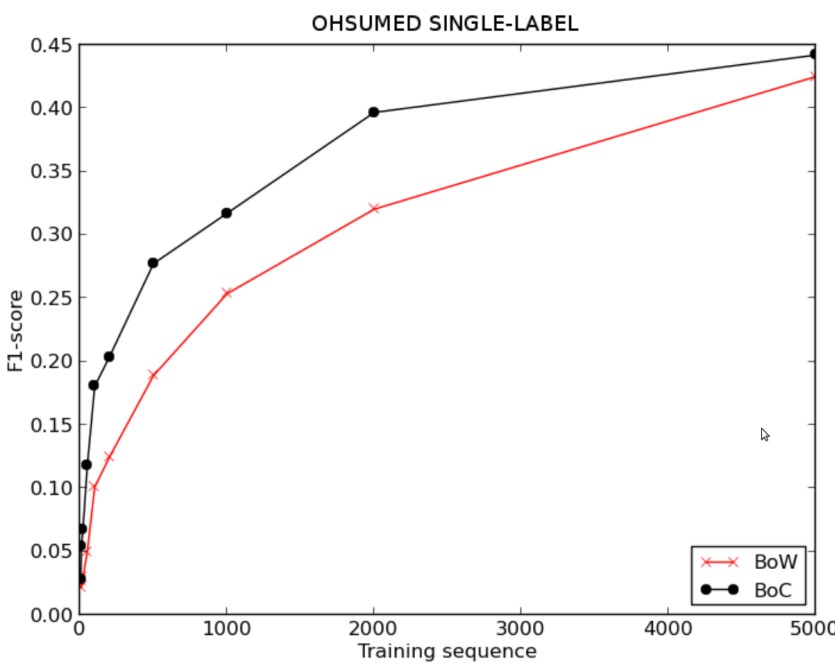

**Figure 4 F1 score for BoW and BoC varying the length of the training sequence in single-labelled OHSUMED corpus.**

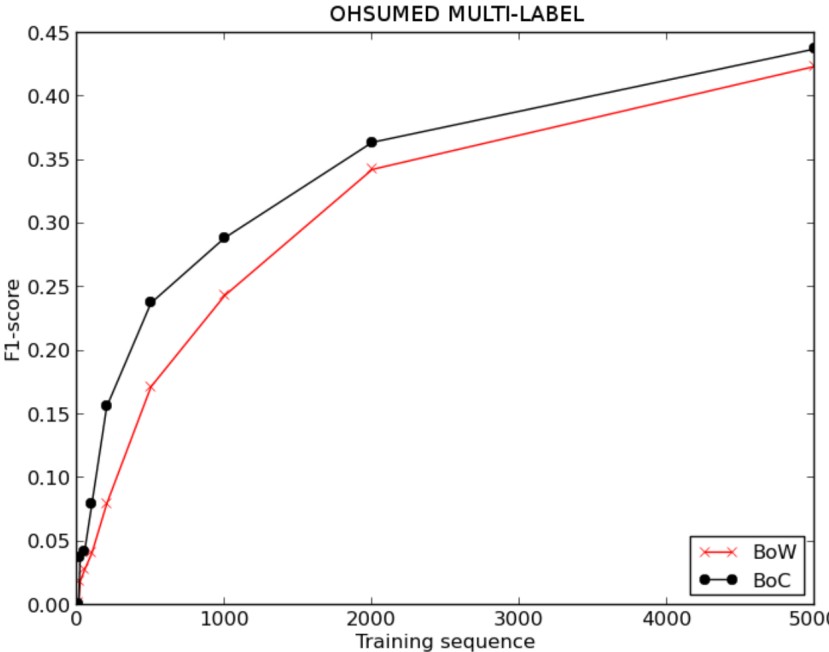

**Figure 5 F1-score for BoW and BoC varying the length of the training sequence in multi-labelled OHSUMED corpus.**

Table 1 **F1-score for BoW and BoC varying the length of the training sequence in single-labelled OHSUMED corpus.**

| | | 5 | 10 | 20 | 50 | 100 | 200 | 500 | 1,000 | 2,000 | 5,000 |
|---|---|---|---|---|---|---|---|---|---|---|---|
| BoW | P | 0.058 | 0.080 | 0.102 | 0.160 | 0.218 | 0.276 | 0.345 | 0.418 | 0.467 | 0.528 |
| | R | 0.129 | 0.074 | 0.106 | 0.163 | 0.248 | 0.307 | 0.377 | 0.426 | 0.471 | 0.519 |
| | F1 | 0.027 | 0.021 | 0.030 | 0.050 | 0.101 | 0.125 | 0.189 | 0.254 | 0.320 | 0.425 |
| BoC | P | 0.089 | 0.134 | 0.213 | 0.281 | 0.308 | 0.332 | 0.421 | 0.460 | 0.512 | 0.535 |
| | R | 0.078 | 0.151 | 0.173 | 0.237 | 0.309 | 0.355 | 0.421 | 0.470 | 0.502 | 0.535 |
| | F1 | 0.029 | 0.054 | 0.067 | 0.118 | 0.181 | 0.204 | 0.277 | 0.317 | 0.397 | 0.442 |

Table 2 **Hamming loss, precision, accuracy, recall and F1-score for BoW and BoC varying the length of the training sequence in multi-labelled OHSUMED corpus.**

| | | 5 | 10 | 20 | 50 | 100 | 200 | 500 | 1,000 | 2,000 | 5,000 |
|---|---|---|---|---|---|---|---|---|---|---|---|
| BoW | HL | 0.061 | 0.061 | 0.063 | 0.063 | 0.062 | 0.062 | 0.059 | 0.058 | 0.056 | 0.054 |
| | P | 0.180 | 0.063 | 0.147 | 0.300 | 0.380 | 0.461 | 0.507 | 0.532 | 0.560 | 0.571 |
| | A | 0.000 | 0.000 | 0.016 | 0.026 | 0.030 | 0.049 | 0.121 | 0.156 | 0.198 | 0.192 |
| | R | 0.001 | 0.002 | 0.031 | 0.047 | 0.056 | 0.082 | 0.200 | 0.288 | 0.385 | 0.482 |
| | F1 | 0.001 | 0.002 | 0.019 | 0.028 | 0.041 | 0.080 | 0.172 | 0.244 | 0.343 | 0.424 |
| BoC | HL | 0.061 | 0.061 | 0.060 | 0.060 | 0.059 | 0.058 | 0.057 | 0.057 | 0.056 | 0.051 |
| | P | 0.021 | 0.063 | 0.300 | 0.415 | 0.457 | 0.526 | 0.543 | 0.553 | 0.556 | 0.591 |
| | A | 0.000 | 0.000 | .0033 | 0.060 | 0.077 | 0.111 | 0.148 | 0.171 | 0.184 | 0.202 |
| | R | 0.001 | 0.001 | 0.054 | 0.085 | 0.121 | 0.182 | 0.273 | 0.340 | 0.404 | 0.481 |
| | F1 | 0.001 | 0.001 | 0.038 | 0.042 | 0.080 | 0.156 | 0.238 | 0.289 | 0.364 | 0.438 |

Table 3 **F1-score for BoW and BoC varying the length of the training sequence in single-labelled UVigoMED corpus.**

| | | 5 | 10 | 20 | 50 | 100 | 200 | 500 | 1,000 | 2,000 | 5,000 |
|---|---|---|---|---|---|---|---|---|---|---|---|
| BoW | P | 0.059 | 0.122 | 0.102 | 0.116 | 0.179 | 0.276 | 0.377 | 0.460 | 0.518 | 0.629 |
| | R | 0.060 | 0.074 | 0.097 | 0.150 | 0.183 | 0.272 | 0.397 | 0.457 | 0.511 | 0.631 |
| | F1 | 0.026 | 0.027 | 0.035 | 0.061 | 0.084 | 0.136 | 0.220 | 0.283 | 0.360 | 0.421 |
| BoC | P | 0.095 | 0.222 | 0.284 | 0.259 | 0.308 | 0.436 | 0.500 | 0.544 | 0.586 | 0.594 |
| | R | 0.049 | 0.093 | 0.148 | 0.247 | 0.321 | 0.432 | 0.515 | 0.557 | 0.590 | 0.598 |
| | F1 | 0.017 | 0.040 | 0.078 | 0.116 | 0.179 | 0.269 | 0.331 | 0.390 | 0.430 | 0.467 |

## DISCUSSION

The results presented in the previous section clearly show the increase in performance of a SVM classifier for categorising biomedical literature when using a Wikipedia-based bag-of-concepts document representation instead of the classical representation based-on-words. It is worth noting that, as can be seen in Fig. 8, the highest increases occur when training sequences are short, because, with enough data, the problems of synonymy and polysemy are masked, and surface overlap performs well.

The increase in classifiers' performance yields important benefits for users—fundamentally medical staff, researchers and students—since a suitable and correct
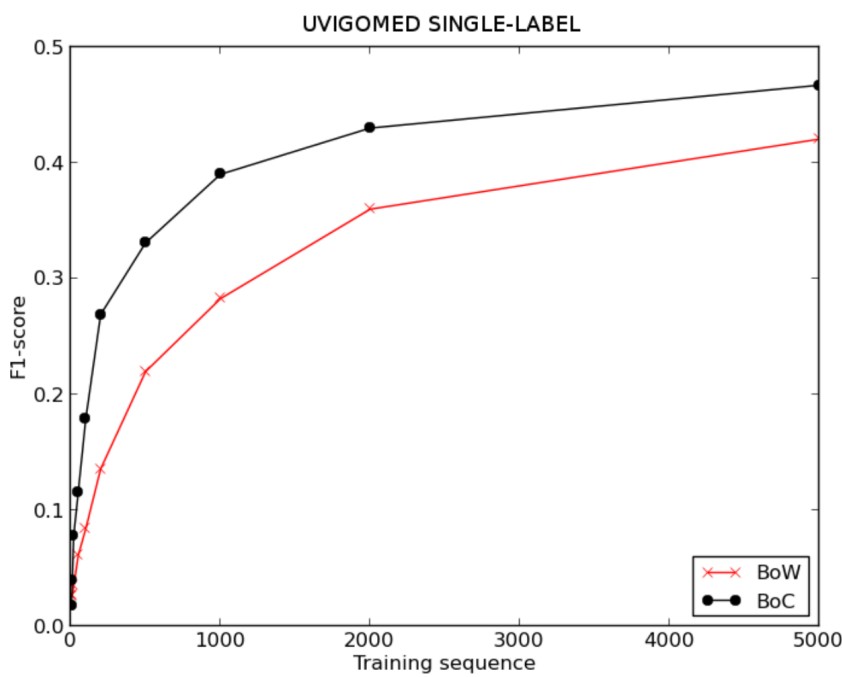

**Figure 6 F1 score for BoW and BoC varying the length of the training sequence in single-labelled UVigoMED corpus.**

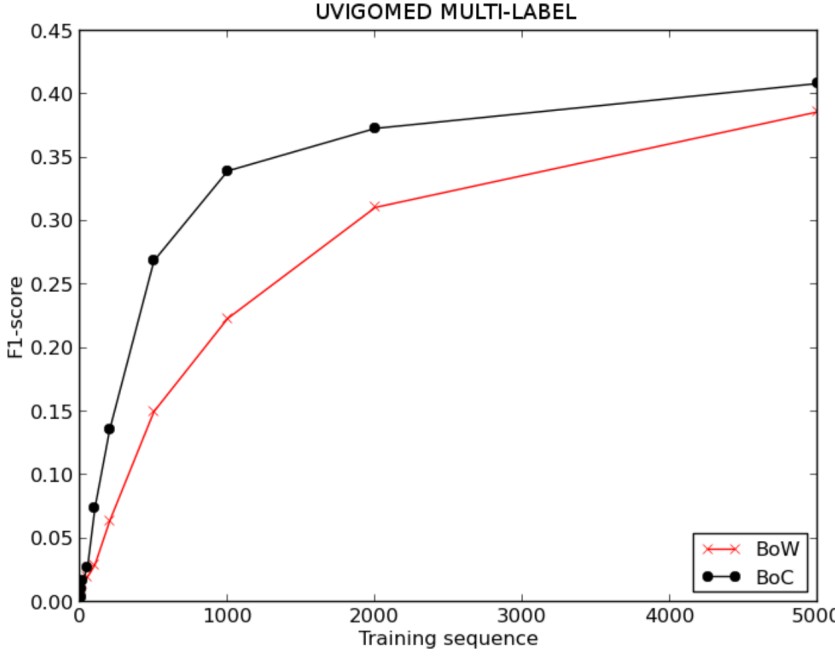

**Figure 7 F1-score for BoW and BoC varying the length of the training sequence in multi-labelled UVigoMED corpus.**

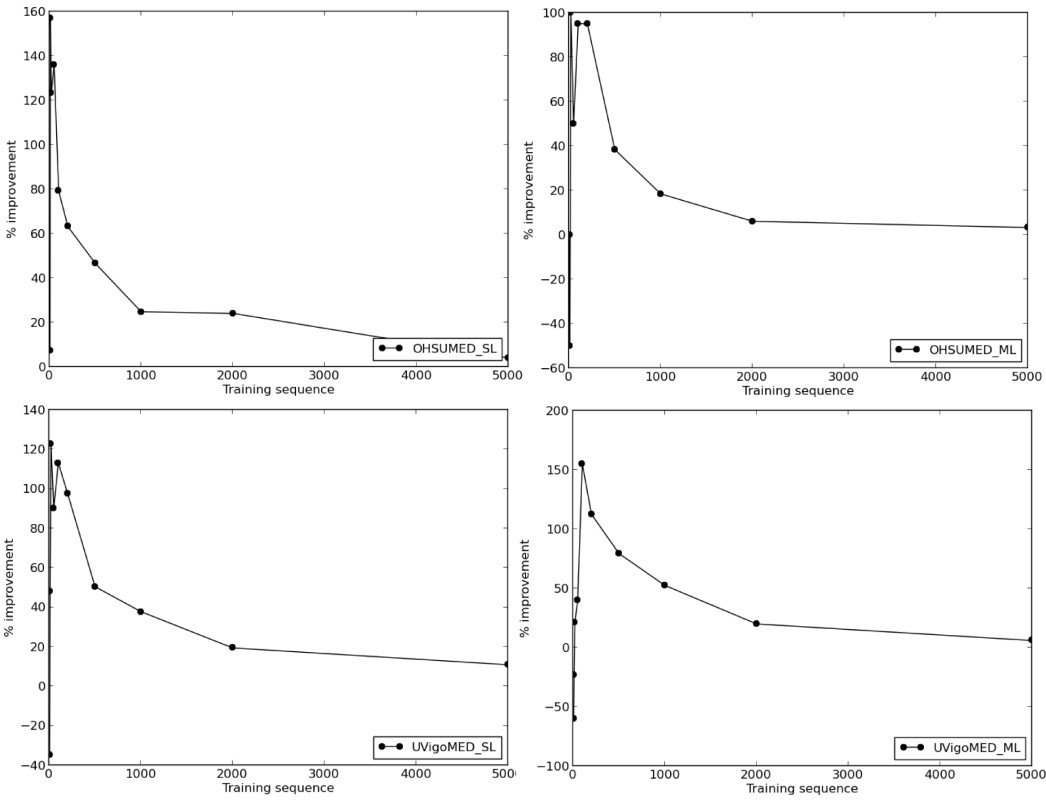

**Figure 8 F1-score percentage improvement for single-labelled OHSUMED, multi-labelled OHSUMED, single-labelled UVigoMED and multi-labelled UVigoMED according to training sequence length variation.**

**Table 4 Hamming loss, precision, accuracy, recall and F1-score for BoW and BoC varying the length of the training sequence in multi-labelled UVigoMED corpus.**

|     |     | 5 | 10 | 20 | 50 | 100 | 200 | 500 | 1,000 | 2,000 | 5,000 |
|-----|-----|-------|-------|-------|-------|-------|-------|-------|-------|-------|-------|
| BoW | HL | 0.087 | 0.090 | 0.068 | 0.064 | 0.065 | 0.065 | 0.061 | 0.059 | 0.056 | 0.055 |
|     | P  | 0.069 | 0.033 | 0.114 | 0.107 | 0.160 | 0.328 | 0.489 | 0.544 | 0.573 | 0.568 |
|     | A  | 0.001 | 0.010 | 0.090 | 0.011 | 0.019 | 0.034 | 0.083 | 0.136 | 0.182 | 0.207 |
|     | R  | 0.024 | 0.031 | 0.015 | 0.016 | 0.029 | 0.062 | 0.152 | 0.229 | 0.312 | 0.414 |
|     | F1 | 0.010 | 0.013 | 0.014 | 0.020 | 0.029 | 0.064 | 0.150 | 0.223 | 0.311 | 0.386 |
| BoC | HL | 0.086 | 0.071 | 0.062 | 0.061 | 0.060 | 0.060 | 0.056 | 0.054 | 0.052 | 0.052 |
|     | P  | 0.001 | 0.140 | 0.225 | 0.186 | 0.411 | 0.536 | 0.589 | 0.601 | 0.606 | 0.599 |
|     | A  | 0.022 | 0.014 | 0.009 | 0.017 | 0.043 | 0.081 | 0.156 | 0.199 | 0.217 | 0.237 |
|     | R  | 0.021 | 0.021 | 0.014 | 0.023 | 0.069 | 0.138 | 0.282 | 0.364 | 0.414 | 0.454 |
|     | F1 | 0.004 | 0.010 | 0.017 | 0.028 | 0.074 | 0.136 | 0.269 | 0.340 | 0.373 | 0.409 |

categorisation facilitates access to those biomedical articles that are really of interest, thus reducing the time needed to find them.

Comparing the proposed approach to other similar approaches in the literature is not an easy task, due to the lack of biomedical literature classification systems that

use a general-purpose semantic annotator, the variety of corpora—and subsets of them—employed, the variety of classification algorithms employed, and the different performance measures used. The only work that uses a general-purpose semantic annotator to classify biomedical literature is *Huang & Milne (2012)*, who classify a subset of MEDLINE—Med100, without specifying whether it is single or multi-label—using a KNN algorithm, and with a proportion of training documents similar to our work (83%). The authors report a F1-score—they do not specify whether it is macro or micro—about 53%. Regarding the use of domain-specific semantic annotators to create representations of documents in biomedical literature classification tasks, we can cite the work of *Zhou, Zhang & Hu (2008b)*, where the authors classify, using a Naïve Bayes algorithm, a subset of OHSUMED corpus comprising only 7,400 documents of the year 1991 that belong to just one category from a total of 14—instead of the 23 that comprises the original OHSUMED corpus—obtaining a macro F1-score of 64%, and using as training sequence 33% of documents of the corpus; and *Yetisgen-Yildiz & Pratt (2005)*, where the authors use an SVM to classify a non-standard subset of OHSUMED corpus composed of 179,796 titles of biomedical articles belonging to 1,928 MeSH categories—without specifying if it is single or multi-label—providing micro F1-score values of 57%.

Finally, the study leaves open lines to future research. The work presented in this paper may be extended by applying the classifier and document representation proposed to the classification of medical histories and patient records, using the proposed document representation along with other classification strategies and algorithms, experimenting with other semantic annotators, and conducting more experiments with other corpora. Another possible future line is the design and development of a software application that allows the visualisation of the documents classified according to the proposal presented in this paper. Thus, users may interact with the application and perform exploratory searches through the categories in which documents are classified. In addition, this will allow us to receive input from users about the results of the classification.

## CONCLUSIONS

This study presents the benefits of using a Wikipedia-based bag-of-concepts document representation and its application to the SVM classification algorithm to classify biomedical literature into a predefined set of categories. The experiments conducted showed that the BoC representation outperforms the classical BoW representation by up to 157% for the single-label problem and up to 100% for the multi-label problem for the OHSUMED corpus. In addition, we created a purpose-built corpus—UVigoMED—that comprises biomedical articles belonging to MEDLINE of year 2014, in which the performance of the classifier using the BoC representation outperforms BoW by up to 122% for the single-label problem and up to 155% in the multi-label problem.

In consequence, we conclude that a Wikipedia-based bag-of-concepts document representation is superior to a baseline BoW representation when it comes to classifying biomedical literature. This is especially true when training sequences are short.

### Funding

Research partially supported by the Galician Regional Government under project GRC2013-006 (Consolidation of Research Units) and through REDPLIR (Red Gallega de Procesamiento del Lenguaje y Recuperacion de Informacion)—R2014/034. The funders had no role in study design, data collection and analysis, decision to publish, or preparation of the manuscript.

### Grant Disclosures

The following grant information was disclosed by the authors:
Galician Regional Government: GRC2013-006.
REDPLIR (Red Gallega de Procesamiento del Lenguaje y Recuperacion de Informacion): R2014/034.

### Competing Interests

The authors declare there are no competing interests.

### Author Contributions

- Marcos Antonio Mouriño García and Roberto Pérez Rodríguez conceived and designed the experiments, performed the experiments, analyzed the data, contributed reagents/materials/analysis tools, wrote the paper, prepared figures and/or tables, reviewed drafts of the paper.
- Luis E. Anido Rifón contributed reagents/materials/analysis tools, reviewed drafts of the paper.

### Data Availability

UVigoMED Corpus: http://itec-sde.net/UVigoMED.zip.

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
