# Peer review of "Biomedical literature classification using encyclopedic knowledge: a Wikipedia-based bag-of-concepts approach"

_PeerJ, doi:10.7717/peerj.1279_

## Round 0.1 · original submission · Minor Revisions

Look forward to receiving your revision

George

·

Basic reporting

Very thorough and well-considered paper.

Experimental design

It was troubling to me that the authors used "scraping techniques" to capture MEDLINE content. As far as I know, they could have simply downloaded all articles and metadata for MEDLINE 2014. The fact that they created their own scaping program, makes one worry that, if they made the smallest mistake, all of their UVigoMed results could be incorrect due to raw-data issues. I would strongly recommend that they simply download all 2014 MEDLINE articles & metadata and rerun their experiments to make sure they get the same results as reported in the current version of the manuscript.

Validity of the findings

The authors do a good job of motivating their work early by saying, "Biomedical staff and researchers have to deal with a lot of literature in their daily activities, so it would be useful a system that allows for accessing to documents of interest in a simple and effective way."

A real opportunity was missed, in my opinion, not to get input from such people (i.e. the colleagues of the authors) on the results of the BoC approach. What would have been especially powerful is to have taken the instances where the BoC was not in agreement with the NLM's disease categorization for a given article and to ask them whether they felt that BoC categorization was better or worse than the NLM categorization. In other words, what if this new approach is preferred over the official MEDLINE categorization by Vigo biomedical-research and clinical practitioners for some small subset of papers? That would be truly groundbreaking.

If the authors do not want to try to add this to the current analysis, I would suggest that they at least add it to the Discussion as part of a Future-Directions subsection.

Reviewer 2 ·

Basic reporting

With regards to literature review, the authors should spend more time outlining the specifics regarding the suboptimal nature of the BoW model. Little time is spent here even though it is the foundation of the study. "Suboptimal" does not provide much information.

By contrast, the Background section (121) seems superfluous. More time should be spent on the BoW and less on extraneous ways of doing BoC, unless the authors can link this section to the experiment more clearly.

Experimental design

Regarding the experimental design, this section is unclear: (336-343)". How does the creation of the BoC compare to the process for creating or obtaining BoW?

Validity of the findings

No issues

Additional comments

The article uses a casual style with less precision than might conform to professional standards of expression. The reader often has to go back to reread due to poor word use and sentence structure (see lines 75-78 for an example). Indicating that "several experiments" were conducted (230-231) is too imprecise for experimental language. A quick review of the piece by a colleague should find the errors in syntax, and the more casual or imprecise uses of language can be addressed by the author.

---

## Round 0.2 · accepted · Accept

Thank you for addressing the issues raised